# Target product profiles of laboratory and data analytical frameworks for genotyping to monitor antimalarial efficacy

Mateusz M. Plucinski[1☉], Amy Wesolowski[2☉], Inna Gerlovina[3], Aimee R. Taylor[4], Jessica Briggs[3], Andrés Aranda-Díaz[5], Monica Golumbeanu[6,7], Marko Bajic[1], Jeffrey A. Bailey[8], Joel L. N. Barratt[9], Caroline Buckee[10], Awa B. Deme[11], Ingrid Felger[12], Anita Ghansah[13], Ian Hastings[14], Johanna Helena Kattenberg[15], Alfredo Mayor[5,16], Didier Menard[17,18,19,20], Leah F. Moriarty[1], Daniel Neafsey[10,21], Lucy Okell[22], Isabella Oyier[23], Jaishree Raman[24,25,26], Philip J. Rosenthal[3], Anna Rosanas-Urgell[15], Robert Verity[22], Sarah K. Volkman[10,21,27], Christian Nsanzabana[6,7‡], Bryan Greenhouse[3‡*]

1 Malaria Branch, Centers for Disease Control and Prevention, Atlanta, Georgia, United States of America, 2 Department of Epidemiology, Johns Hopkins Bloomberg School of Public Health, Baltimore, Maryland, United States of America, 3 Division of HIV, ID, and Global Medicine, Department of Medicine, University of California San Francisco, San Francisco, California, United States of America, 4 Department of Global Health, Institut Pasteur, Université Paris Cité, Paris, France, 5 ISGlobal, Barcelona, Spain, 6 Department of Epidemiology and Public Health, Swiss Tropical and Public Health Institute, Allschwil, Switzerland, 7 University of Basel, Basel, Switzerland, 8 Department of Pathology and Laboratory Medicine, Brown University, Providence, Rhode Island, United States of America, 9 Department of Pathology and Laboratory Medicine, Emory University School of Medicine, Atlanta, Georgia, United States of America, 10 Department of Immunology and Infectious Disease, Harvard TH Chan School of Public Health, Boston, Massachusetts, United States of America, 11 Centre International de Recherche et de Formation en Genomique Appliquee et de Surveillance Sanitaire, Dakar, Senegal, 12 Department of Medical Parasitology and Infection Biology, Swiss Tropical and Public Health Institute, Allschwil, Switzerland, 13 Department of Parasitology, Noguchi Memorial Institute for Medical Research, College of Health Sciences, University of Ghana, Legon, Ghana, 14 Department of Tropical Disease Biology, Liverpool School of Tropical Medicine, Liverpool, United Kingdom, 15 Department of Biomedical Sciences, Institute of Tropical Medicine, Antwerp, Belgium, 16 Centro de Investigação em Saúde de Manhiça (CISM), Maputo, Mozambique, 17 Malaria Genetics and Resistance Team (MEGATEAM), UR 3073 - Pathogens Host Arthropods Vectors Interactions Unit, Université de Strasbourg, Strasbourg, France, 18 Malaria Parasite Biology and Vaccines, INSERM Unit 1347 – ParasitInnov, Institut Pasteur, Université Paris Cité, Paris, France, 19 Laboratory of Parasitology and Medical Mycology, CHU Strasbourg, Strasbourg, France, 20 Senior Chair program, Institut Universitaire de France (IUF), Paris, France, 21 Infectious Disease and Microbiome Program, Broad Institute, Cambridge, Massachusetts, United States of America, 22 MRC Centre for Global Infectious Disease Analysis, Imperial College London, London, United Kingdom, 23 Biosciences Department, KEMRI-Wellcome Trust Research Programme, Kilifi, Kenya, 24 Centre for Emerging Zoonotic and Parasitic Diseases, National Institute for Communicable Diseases, Johannesburg, South Africa, 25 Wits Research Institute for Malaria, University of Witwatersrand, Johannesburg, South Africa, 26 UP Institute for Sustainable Malaria Control, University of Pretoria, Pretoria, South Africa, 27 School of Nursing, Simmons University, Boston, Massachusetts, United States of America

☉ These authors contributed equally to this work.
‡ CN and BG also contributed equally to this work.
* bryan.greenhouse@ucsf.edu

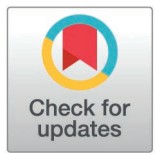

## Abstract

Therapeutic efficacy studies (TESs) are the standard to evaluate antimalarial drug efficacy and guide malaria treatment policy. TESs are particularly relevant now, with resistance to first-line regimens emerging in sub-Saharan Africa. For TESs, a range of parasite genotyping and data analyses are available for genotype correction, a

**Data availability statement:** All data are included.

**Funding:** This work was supported by the US National Institutes of Health (K24AI144048 to BG and U01AI184646 to BG and AM), the Gates foundation (INV-081860 to JB and BG, INV-037564 to AG, INV-067310 to AM, AAD, and BG, INV-009416 to DN, and INV-049909 to SKV), the Belgian Directorate-General for Development Cooperation (DGD) to JHK and ARU, the Institut Universitaire de France (IUF) under the Senior Chair program (2024–2029) to DM and the Fondation pour la Recherche Médicale (FRM) under the 'Équipes FRM 2024' program (Grant Agreement No. EQU202403018026 to DM), the European Union (101110393 to ART). LO and RV acknowledge funding from the MRC Centre for Global Infectious Disease Analysis (reference MR/R015600/1), jointly funded by the UK Medical Research Council (MRC) and the UK Foreign, Commonwealth & Development Office (FCDO), under the MRC/FCDO Concordat agreement and is also part of the EDCTP2 programme supported by the European Union, and funding by Community Jameel. The funders had no role in study design, data collection and analysis, decision to publish, or preparation of the manuscript.

**Competing interests:** The authors have declared that no competing interests exist.

process to distinguish whether recurrent parasitemia after therapy is due to recrudescence of initially infecting parasites (treatment failure) or a new infection. The choice of methods for laboratory genotyping and data analyses can have a large effect on how outcomes are classified, and thereby on trial results. The currently recommended and most widely used laboratory and analytical methods for TES genotyping do not incorporate recent methodological advances and can produce biased results. As such current TES results can be difficult to interpret, especially in areas with high malaria transmission, such as much of sub-Saharan Africa. Thus, improving the accuracy and reliability of TES genotyping and data analysis are a major priority. To that end, we present target product profiles that outline key specifications for genetic data generation, processing, and data analysis, with the goal of creating rigorous and consistent community standards. Primary recommended specifications for laboratory methods include high sensitivity, specificity, and reproducibility, and guidance on the number and genetic diversity of targets; criteria which are best and likely only met by amplicon sequencing. Primary recommendations for data analysis methods include high classification accuracy, accounting for errors in genotyping, and accounting for alleles matching by chance. All laboratory and data analysis methods used should be systematically validated and publicly documented so that TES results, which have major policy implications, can be relied upon for sound programmatic decision making.

## Introduction

The ability to treat and cure *Plasmodium falciparum* malaria constitutes a centuries-long arms race between the development of new antimalarial compounds and the malaria parasites' evolution to resist these drugs. The most recent class of antimalarials, the highly efficacious artemisinin-based combination therapies (ACTs), has contributed to a dramatic decline in the malaria burden over the past 20 years [1]. However, partial resistance to artemisinin derivatives in combination with resistance to some partner drugs resulted in therapeutic clinical failures in Southeast Asia and now threatens gains made in sub-Saharan Africa, where the malaria burden is highest [2].

Therapeutic efficacy studies (TESs) are the primary means by which antimalarial drug efficacy is monitored under programmatic conditions and choices of therapy are determined [3]. In these trials, patients with microscopy-confirmed *P. falciparum* infection are treated with antimalarials and followed for 4–6 weeks to assess clinical and parasitological responses to treatment. Blood samples are collected from patients on the day they enroll, before treatment (Day 0), and routinely for the remainder of the follow-up period. Treatment failure is defined as slow or incomplete clearance in the three days immediately following treatment, or recurrence of parasitemia during follow-up despite initial clearance of detectable parasitemia.

Antimalarial trials are complicated by the possibility, especially in high transmission settings, of new blood-stage infections acquired via ongoing exposure to infectious mosquito bites and subsequent sporozoite inoculation during follow-up, which can confound assessment of treatment efficacy. Thus, cases of recurrent *P. falciparum* parasitemia can be due to either (a) recrudescence, *i.e.*, true treatment failure, (b) new infection, *i.e.*, an infection developing from a new mosquito inoculation, or (c) a mixture of both recrudescent and newly inoculated clones. Genotype correction (also known as molecular correction) is performed to distinguish between these situations [3,4]. Paired samples from the same patient are collected on Day 0 and the day of recurrence of microscopically detectable parasites. Parasites from both samples are genotyped and compared to distinguish recrudescence from new infection (or a mixture of both), with matched genotypes implying recrudescence. In the corrected efficacy estimate - the primary outcome of the TES - new infections are not considered treatment failures and do not contribute to the failure rate. While genotype correction is straightforward in principle, TES outcomes are often classified based on subjective and unreliable genotyping methods, as well as ad-hoc, non-standardized data analysis, leading to potentially biased and inconsistent results [2,5–11].

Distinguishing recrudescence events from new infections in high transmission areas poses particular challenges due to high rates of recurrent parasitemia and a high prevalence of infections containing multiple genetically distinct clones (polyclonal infections) [6,7]. Characterization of a limited number of length polymorphic loci (*e.g.*, merozoite surface proteins and microsatellites) using gel or capillary electrophoresis has been historically used as the standard for distinguishing recrudescence from new infection [12]. However, the discriminatory power of these loci can be limited in high transmission settings, in part due to limited analytical sensitivity to detect minority alleles and difficulty in generating and reproducibly analyzing length polymorphism data (S1 Table). In response, recent (2021) World Health Organization (WHO) guidelines advocate for the integration of amplicon deep sequencing and probabilistic modeling to enhance the diagnostic accuracy of genotype correction, especially in high-transmission settings where traditional genotyping methods may be particularly challenging [13].

Even with accurate genotyping, comparison of genotypes in paired samples is complex. Data analysis methods that perform such comparisons can be broadly divided into two approaches: heuristic, rule-based approaches and statistical approaches, which are either based on a probabilistic model or on hypothesis testing using empirical distributions [14,15]. The most common rule-based approach is match counting, where a recrudescence is defined when the number of loci for which shared alleles are detected in both samples exceeds a predefined threshold. The current WHO standard, dating back to 2007, is to use a strict 3/3 match counting approach [12]. In this approach, three length polymorphisms are genotyped and the infection is classified as recrudescent if at least one allele of each locus matches between samples from Day 0 and the day of recurrence. Rule-based approaches like these are subject to potential biases. For example, new infections might be misclassified as recrudescent in low transmission areas due to an increased chance of allele matching resulting from low allelic diversity, in particular if only a limited number of loci are included. In high transmission scenarios, many study participants will carry polyclonal infections, *i.e.,* the complexity of infection (COI) will be greater than 1. The presence of multiple circulating clones can bias results toward recrudescence due to the increased number of alleles from a finite allele pool that are being compared. At the same time, high-parasitemia clones in polyclonal infections can mask lower-parasitemia clones, resulting in undetected alleles and biasing results away from recrudescence. In all settings, biological constraints (such as organ tissue sequestration of clones) and technical limitations (such as PCR amplification bias) can lead to misclassification of recurrent parasitemia [4,16–19]. Unlike rule-based approaches, statistical approaches have been designed to incorporate factors such as parasite population genetic diversity, polyclonal infections, undetected alleles, and other factors while also providing the ability to express uncertainty of the classification [14,15,20,21]. These statistical approaches have not yet been widely implemented in TESs.

With the multiple limitations in recommended genotype correction methods for TES (laboratory and data analysis), few available antimalarials to choose from, and no new drugs expected soon, policymakers find themselves needing to make difficult choices with a lack of robust evidence. Underestimating recrudescence can result in continued use of a failing

drug, worsening patient outcomes and providing continued selective pressure for resistance. Overestimating recrudescence can result in premature recall of an effective drug, when the availability of choices may be limited and the cost of changing policy substantial. In both cases, incorrect policy decisions will lead to mismanagement of patients, inadequate mitigation of resistance, and sub-optimal use of limited public resources. Consequently, improving the accuracy and reliability of genotype correction for TESs is a major priority for the malaria community. To that end, we present target product profiles (TPPs) that outline key specifications for genetic data generation and processing (*i.e.*, from blood sample to allele calls) and the statistical analysis of these data (*i.e.*, from allele calls to classification and overall study results). The target audience for this document includes national malaria programs, researchers, funders and multilateral organizations who all work together to inform and guide malaria case management policy decisions. Given the likelihood of continued evolution of methods, the TPPs intentionally do not specify that any particular genotyping or data analysis method be used, with the objective that any method rigorously validated to meet these criteria should provide the desired output. However, specific methods are occasionally referenced for clarity or to provide examples.

## Methods

Draft TPPs for genotyping assays and statistical analysis frameworks for genotype correction of antimalarial TES were developed with characteristics described as either 'minimal' or 'optimal'. Minimal was defined as the basic characteristics that should be met to produce reliable results for a TES. Optimal was defined as any additional desired characteristics beyond minimal. Experts were selected based on their experience and expertise in the field of genotyping analyses of malaria parasites, including from laboratory and data analysis perspectives. All experts who provided feedback are coauthors on this manuscript or mentioned by name in the acknowledgements and are aware their feedback was incorporated in the drafting of this manuscript. An initial group of experts was convened to discuss and outline the two TPPs: one on laboratory genotyping and the other on data analysis. We defined the scope of the laboratory assay TPP as starting from DNA extraction and ending at defining alleles present in the blood sample, including any data processing or bioinformatics needed to obtain allele calls. We defined the scope of the data analysis TPP as starting from allele calls and ending at final genotype correction results, *i.e.,* whether the infections in participants with recurrent parasitemia were more likely a recrudescence or new infection. Subsequently, experts in the field were engaged to provide additional comments by broadly soliciting feedback on a volunteer basis, with most of these experts focusing on either the laboratory or data analysis TPP based on their expertise.

## Simulations

We performed simple simulations to provide practical guidelines on the number and diversity of genetic loci required to distinguish recrudescence from new infection at a level of accuracy listed in the TPP (S2 Table). Details of the simulations including code and results are provided. In brief, we used a grid of approximately 4000 (2^12) heterozygosity (He) values with corresponding randomly generated population allele frequencies; these frequencies were repeated for each locus. For each constructed panel and each complexity of infection (COI) value, we simulated 10,000 Day 0 samples and 10,000 recurrent samples representing new infections. A second set of recurrent samples representing recrudescences was constructed by replacing one of the newly infecting strains with a strain from the corresponding Day 0 infection. Genotyping errors were added to each sample, assuming 90% probability of allele detection and a false positive rate of 2%. We then used the *asterTES* package to classify each Day 0 – Day of recurrence pair as a recrudescence or a new infection [20]. Classification results of new infection pairs were used to assess specificity and classification results of recrudescent pairs to assess sensitivity. The process was repeated until the desired performance characteristics were achieved (95% specificity and 95% sensitivity). More details and simulation code are included in the supplemental materials. Classification results were then used to calculate specificity and sensitivity; the smallest He value to achieve the desired performance level is presented. Code for the simulations is included with the supplemental materials.

## Results

**Participants**: In total, ten individuals contributed to the initial drafting of the TPPs (first eight and last two authors), followed by 19 additional experts assisting with edits and comments. All the experts are engaged in genetic analyses of malaria parasites, with expertise in *P. falciparum*. Participants were primarily based in Europe and the USA, with 3 participants from Africa. Nineteen participants focused primarily on the laboratory component while ten participants focused on the data analysis component. There was an even breakdown for sex (16/29 female), and all had an advanced degree (PhD or MD).

## General characteristics

A distillation of key messages from this exercise, including definitions, current methods and their limitations, and proposed solutions is presented in Table 1. The detailed TPP for laboratory genotyping is outlined in Tables 2–4 and for statistical analysis in Tables 5 and 6.

Intended use: The overall goal of genotype correction in a TES is to distinguish new infections from recrudescent infections. To this end, the goal of the laboratory assay is to generate allele calls. The goal of the statistical analysis framework is to use these allele calls on pairs of samples from the same individual to robustly classify recurrence as a new infection or recrudescence, or alternatively assign an accurate probability for each recurrence classification.

Once individual-level classifications are available, the final analytic step in a trial is to calculate the overall study-level failure rate. Optimally, uncertainty from allele calls through recurrence classification into study-level failure rates would be

**Table 1. Summary of current methods and proposed improvements to laboratory assays and data analysis methods for molecular correction.**

|  | Laboratory assay | Data analysis |
|---|---|---|
| Objective | Generate genotype data from parasites from paired baseline and recurrent parasitemia samples with enough sensitivity to detect all strains that persist from baseline to day of recurrence and enough resolution to allow differentiation of unrelated strains | Analyze paired genotype data to accurately classify cases of recurrent parasitemia as either due to recrudescence or new infection |
| Scope | Wet lab assays that generate raw genetic data and bioinformatics pipelines that identify the alleles present from these data. | Algorithms that use data on alleles present to infer recrudescence or new infection state for cases of recurrent parasitemia; optional extension into statistical algorithms that also generate study arm-level failure rates |
| Current guidance | Capillary electrophoresis of three length polymorphic markers | Match-counting algorithm where recrudescence is defined by a shared allele at all loci |
| Key limitations of current methods | Poor sensitivity in detecting minor alleles; limited diversity in some settings; low accuracy; limited reproducibility | No consideration of missingness or noise in genotyping data; no incorporation of number or frequency of alleles detected, i.e., attempt to account for random matching; failure to estimate uncertainty in classification |
| Proposed way forward | Assess suitability of newly developed or existing genotyping methods using criteria laid out in Tables 2–4; Validate genotyping methods using experimental approaches and/or comparison against other methods | Design new or adapt existing statistical frameworks and verify they comply with criteria laid out in Tables 5 and 6; Validate and compare statistical algorithms against each other using *in silico* experiments |

**Table 2. General scope and characteristics of the genotype correction laboratory assay framework.**

| Characteristic | Minimal Requirements | Optimal Requirements | Comments |
|---|---|---|---|
| Intended use | Genotype correction to distinguish recrudescence from new infection | Genotype correction to distinguish recrudescence from new infection Same assay evaluates validated (+/- candidate) molecular markers of drug resistance | Accurate assessment of intermediate measures, such as complexity of infection and allele frequency, are likely necessary to properly evaluate outcomes as discussed in the data analysis TPP. |
| Target end users | Well-equipped laboratories for required methods with highly qualified staff Genotyping "hubs" in malaria endemic regions; could potentially include private sector | National laboratories or research laboratories in malaria endemic countries | Minimal and optimal here defined from the standpoint of the assay, not its implementation; an optimal assay would require less in the way of infrastructure than a minimal assay. |
| Target population | Malaria patients enrolled in antimalarial drug clinical trials | Same as minimal | |
| Application setting | All laboratories with the required equipment and highly trained staff | Laboratories with limited equipment and staff with basic molecular training | Minimal and optimal here defined from the standpoint of the assay, not its implementation; an optimal assay would require less physical and personnel infrastructure than a minimal assay. |
| Access and usability | Required infrastructure and equipment should be available in regional genotyping hubs in malaria endemic regions | Required infrastructure and equipment should be available in each malaria endemic country performing trials. | From the standpoint of capacity strengthening, it would be ideal to have each country perform its own genotype correction. This consideration needs to be balanced against the likelihood that having fewer, well-resourced labs (*e.g.,* via a framework to support regional hubs) may be more cost-effective and produce more reproducible results. Capacity strengthening of national labs could benefit from a standardized plan for certification, training, and procurement of reagents which could be facilitated by appropriate organizations such as WHO and Africa CDC. |
| Output | The primary outputs for analysis will be allele calls (*e.g.,* microhaplotype sequences for amplicon sequencing or binned lengths for length polymorphisms). Assays should additionally output raw genetic data (*e.g.,* FASTQ file for sequencing,.fsa files for capillary electrophoresis length polymorphisms). | Minimal plus measure of uncertainty around allele calls (including for molecular markers of drug resistance, if included), and/or within sample allele frequencies | Additional outputs pertaining to classification are addressed in the analysis TPP. |

propagated into the final failure rate calculations. However, an extensive review of the methods for study-level failure rate estimation (*e.g.*, a competing risk approach versus Kaplan–Meier estimator) is outside the scope of this exercise.

Target study population: The target population is malaria patients enrolled in a clinical trial to monitor the efficacy of an antimalarial drug.

Target end users and implementation: The target users may vary based on the logistics of study implementation. For the laboratory assay, users will always include highly qualified staff at well-equipped laboratories for the required methodologies and may include staff at resource-limited laboratories. This includes national public health laboratories, research laboratories, and regional centers of excellence. For the analytical framework, users will include TES data analysts and investigators who will conduct statistical analyses; with appropriate development it should be relatively straightforward to make data analysis widely accessible to users with a range of backgrounds and expertise. To ensure statistical frameworks meet expectations and encourage uptake, algorithms and code should meet the FAIR criteria for research software [22].

**Table 3. Performance characteristics of the genotype correction laboratory assay framework.**

| Characteristic | Minimal Requirements | Optimal Requirements | Comments |
|---|---|---|---|
| **Analytical sensitivity** | Alleles at ≥ 10% within sample allele frequency (WSAF) detectable in at least 95% of samples if clones are from dried blood spots (DBS) with ≥ 100 parasites/μl AND Alleles at ≥ 5% WSAF detectable in at least 95% of samples if clones are from dried blood spots (DBS) with ≥ 500 parasites/μl | Alleles at ≥ 2% WSAF detectable in at least 95% of samples if clones are from DBS with ≥ 100 parasites/μl | Samples from day of recurrence may have parasite densities lower than initial presentation; requirements for sensitivity need to account for these densities. Sensitivity needs to be validated with blinded, mixed-clone control infections replicating sample type and extraction, e.g., for minimal 5% at 100 parasites/μl. An undisclosed subset of mixed controls should contain > 2 clones. Performing evaluation in replicates, *e.g.,* at least triplicate, is suggested. Initial validation of methods should include samples run at least in triplicate. Evaluation includes allele-calling methods, *e.g.,* bioinformatic pipelines for sequencing data. Evaluation of sensitivity, specificity, and reproducibility can best be ensured with an external quality assurance program, ideally coordinated by a third party such as WHO. This includes distributing validation materials including mixed-clone controls as described. For length polymorphism only: Because of preferential amplification of shorter amplicons, the larger amplicon should be selected as the minority allele and be the same allelic family as the major allele for at least a subset of controls (*e.g.,* for msp1: 3D7=minority clone, K1 = dominant clone). |
| **Analytical specificity (detection of alleles)** | 95% of alleles | 99% Estimates of uncertainty of allele calls are provided, *e.g.,* probability of being a true positive allele vs. artifact. | Evaluation includes allele calling methods. Evaluation of sensitivity, specificity, and reproducibility can best be tested with validation in multiple laboratories, and continued performance of individual laboratories ensured with an external quality assurance program, ideally coordinated by a third party. |
| **Reported values** | For sequencing-based methods: Reads per allele per locus per sample (validation and trial samples). Details of bioinformatic pipeline used, including code, version, any user-specified parameters, *i.e.,* enough to fully reproduce the allele call results from the raw data. For length polymorphisms: Presence of genotypes per locus per sample (validation and trial samples). Details of allele calling, *e.g.,* if alleles are binned the strategy employed to select bin width and location, any cutoffs used to distinguish alleles from artifact. This should be enough information to fully reproduce the allele call results from the raw data. | For sequencing-based methods: minimal plus probability of allele being a true positive (*i.e.,* estimates of uncertainty of allele call from the bioinformatic pipeline). For length polymorphisms: minimal plus estimated WSAF for binned alleles per locus per sample and probability of allele being a true positive (*i.e.,* estimates of uncertainty of allele call). | |

*(Continued)*

## Global Public Health

PLOS

**Table 3.** (Continued)

| Characteristic | Minimal Requirements | Optimal Requirements | Comments |
|---|---|---|---|
| **Number and diversity of loci** | See S2 Table. | Meeting criteria such as calculated in S2 Table using simulations parameterized by setting-specific allele frequencies and distribution of complexity of infection, ensuring that the number and diversity of targets is in excess of what is required per setting. | These are guidelines, but careful attention will need to be paid to analysis outcomes to ensure high enough resolution for specific settings. S2 Table should be considered the first iteration to be updated as additional evidence is generated. Note that in practice a panel may contain a larger number of loci to ensure that the minimal number have sufficient diversity in a given setting and to be robust to missing data, e.g., in low parasitemia samples. |
| **Reproducibility of replicates within a lab** | Controls (DBS with known composition of single and multiple laboratory clones) should be run in triplicate each time the assay is used: 98% of times the user gets the same set of alleles in monoclonal controls, 80% of times the user gets the same set of alleles in polyclonal controls containing minority WSAF of 10% and 2%.<br>At least 10% of study samples (with a focus on paired Day 0 and recurrence samples) should be run in duplicate and should meet the criteria above. | Controls (DBS with known composition of laboratory clones) 99% of times the user gets the same set of alleles in monoclonal controls,<br>90% of times the user gets the same set of alleles in polyclonal controls containing minority WSAF of 10% and 2%.<br>All study samples should be run in triplicate and should meet the criteria above. At least 20% of Day 0 and recurrence sample pairs are validated by an external laboratory and should meet the criteria above. | Evaluation of sensitivity, specificity, and reproducibility can best be tested with validation in multiple laboratories, and continued performance of individual laboratories ensured with an external quality assurance program, ideally coordinated by a third party. |

## Technical and performance characteristics

*Laboratory TPP:* The most important performance criteria were sensitivity in detecting minority clones, specificity of allele calling, number and genetic diversity of targets (including their allelic frequency distributions), and reproducibility of the assay, including replicates between labs. Sensitivity was discussed at length, considering the importance of detecting minor variants at a range of parasitemia levels and the practical limitations of current assays; inadequate sensitivity limits the ability to detect recrudescent clones. Specificity must be sufficient to minimize the probability of false positive alleles resulting in misclassification of a new infection as a recrudescence. Multiple, diverse targets are required to ensure accurate recurrence classification within the population of interest. The number and diversity of targets required may vary considerably depending on COI; thus we performed simulations to create basic guidelines as part of the TPP (Table 4). Ideally, the number and diversity of targets would be sufficient to account for worst case scenarios. Finally, validity of employed assays and their reproducibility on shared samples between labs should be prioritized, for example with the use of an external quality assurance program managed by a third-party organization.

*Data Analytic TPP:* The most important performance criteria include classification of specificity and sensitivity that account for errors in genotyping, particularly non-detection of minority alleles, and alleles matching by chance. Other criteria include that the statistical algorithm is accurate and consistent, *i.e.*, for a given recurrence, the output of the software results in a stable classification (recrudescence or new infection) or stable per-outcome probabilities with precision improving for increasing numbers of loci. A modular and scalable implementation would allow easy adaptation and

PLOS Global Public Health

**Table 4. Operational characteristics of the genotype correction laboratory assay framework.**

| Characteristic | Minimal Requirements | Optimal Requirements | Comments |
|---|---|---|---|
| **Assay format** | Low throughput (*e.g.,* 20 samples) | High throughput (at least 96-well plate or ~90 samples at a time) | |
| **Assay type** | Any assay meeting minimal performance criteria above, which could include NGS-based or capillary electrophoresis | NGS-based assay meeting performance criteria above. | |
| **Sample matrix** | Dried blood spot (DBS)<br>The assay may work on other sample types such as whole blood. | Same as minimal | |
| **Cost per individual sample replicate (*e.g.,* Day 0 sample or control, from DNA extraction to allele generation including, e.g., checking of library quality, sequencing, etc.)** | Less than USD 100 for all reagents and consumables (i.e., not including labor) in a regional hub or reference center | Less than USD 25 for all reagents and consumables in a regional hub or reference center<br>Less than USD 50 for all reagents and consumables in most national laboratories | In many low-resource settings, even the optimal requirements listed here may present a significant barrier. Subsidization mechanisms, pooled procurement models, and availability of shared equipment (*e.g.,* access to local high-throughput sequencers) may further lower cost and improve access. |
| **Wet lab equipment** | Hoods<br>Thermocycler<br>Means of detecting alleles (*e.g.,* capillary electrophoresis or high throughput sequencer)<br>Other equipment as required<br>Dedicated spaces for DNA extraction, mastermix preparation, and handling of post PCR material to minimize contamination. | Hoods<br>Thermocycler<br>NGS Sequencer<br>Automated systems<br>Other equipment as required<br>Dedicated rooms with appropriate airflow for DNA extraction, mastermix preparation, and handling of post PCR material to minimize contamination. | Preferable that the equipment has support of and utility for other lab activities to share cost for purchase, maintenance, training, etc. |
| **Computational infrastructure needed** | For sequencing-based methods: High performance desktop, laptop, or access to cloud-based resource<br>For length polymorphisms: Standard performance desktop or laptop | For sequencing-based methods: Cloud-based resource and high performance laptop are both options | |
| **Assay throughput** | 200 individual samples (or 100 paired Day 0 and Day of recurrence) samples) in 2 months | 400 individual samples (or 200 paired Day 0 and Day of recurrence samples) in 2 weeks | Numbers here are just for time estimates; proper implementation will require a combination of Day 0 and paired samples, negative and positive controls, etc. |

*(Continued)*

**PLOS Global Public Health**

**Table 4.** (Continued)

| Characteristic | Minimal Requirements | Optimal Requirements | Comments |
|---|---|---|---|
| **Training** | 8 weeks for wetlab technician with general laboratory experience (*e.g.,* PCR) but not with the specific type of assay (*e.g.,* targeted deep sequencing)<br>4 weeks for wetlab technician with prior domain-area expertise<br>8 weeks for bioinformatics or analysis (including allele calling) without prior experience<br>4 weeks for bioinformatics or analysis with prior domain-area expertise | 4 weeks for technician without prior experience<br>2 week for technician with prior experience<br>4 weeks for bioinformatics or analysis without prior experience<br>2 weeks for bioinformatics or analysis with prior experience<br>Continuous evaluation and certification | Minimal and optimal here defined from the standpoint of the assay, not implementation; an optimal assay would require less training. Continuous training is required to maintain proficiency. It is worth considering a staff retention policy to retain technicians with experience Evaluation of competence can best be ensured with an external quality assurance program, ideally coordinated by a third party such as WHO. |
| **QA & QC** | Sensitivity, specificity, and reproducibility require internal and external validation using controls and replication as described above. At a minimum, this will require a set of positive controls with mixes of different laboratory strains at different densities diluted in human blood stored in the sample type used for the assay (*e.g.*, DBS) sufficient to assess the sensitivity for detecting minority clones at the levels defined above. | Formal quality assurance (EQA) scheme administered by a third party such as WHO. | |

extension as new data types (*e.g.*, new genotyping techniques) become available. Implementation should follow best practices in version control and software maintenance to ensure long-term usability and community contributions.

## Operational characteristics

*Laboratory TPP*: Optional criteria for assays include the ability to use a range of blood products, including a dried blood spot since these are easy to collect and maintain. Both low- and high-throughput platforms may be suitable, provided they meet the performance criteria outlined above. However, there was a strong preference for deep sequencing-based assays due to high accuracy, reproducibility, and efficiency, as discussed below. Costs should be reasonable and obtain economies of scale depending on the throughput, *i.e.*, if running a larger number of samples the cost per sample should decrease. Standard training from experienced laboratories to ensure high quality data is needed, in addition to standardized quality assurance and quality control (QA & QC). Genotyping methods that employ equipment that are readily usable for other laboratory activities - synergizing shared costs and training - are preferred.

*Data Analytic TPP*: Optional criteria for data analysis include code which is freely available, well commented, provides user-friendly tutorials, and is easy to use. The algorithm should be implementable on a standard laptop and not require high performance computing resources. As well as being robust to differences in data quality, sample size, and genetic diversity, the algorithm should perform well across a range of parasite populations of varying COI from low to high transmission scenarios.

**Table 5. General scope and characteristics of the TES data analysis framework.**

| Characteristic | Minimal Requirements | Optimal Requirements | Comments |
|---|---|---|---|
| Intended use | Framework explicitly designed to analyze genotyping data from participants with recurrent parasitemia to differentiate between recrudescence and new infection. | Same as minimal, with an additional step of providing an estimate of the corrected efficacy rate with corresponding measures of uncertainty. | |
| Target end users | Data analysts and investigators of therapeutic efficacy studies | Same as minimal | |
| Target population | Patients with microscopy-confirmed *P. falciparum* infection enrolled in a therapeutic efficacy study | Same as minimal | |
| Collection of samples for analysis | For study participants with recurrent malaria, paired samples of malaria infections on Day 0 and Day of recurrence. | Minimal plus all Day 0 samples regardless of recurrence. | Inclusion of additional genotyped Day 0 samples allows for more accurate estimates of allele frequency (and Day 0 population structure), which results in more robust estimates in methods that account for these factors. Suboptimally, estimates can be obtained from other studies in geographic and temporal proximity. |
| Implementation setting | National malaria control programs, genotyping hubs in malaria endemic regions, sentinel sites for therapeutic surveillance, research laboratories, or any other setting where antimalarial drug efficacy is monitored and EQA is performed | Same as minimal | |
| Access and usability | The data analytical framework is available as a statistical program/ or script that can be downloaded from a dedicated website or package repository, with straightforward installation and usage instructions. Software should not be operating system specific. Should include a minimally reproducible example dataset to ensure the framework is being correctly implemented. Should have diagnostic messages and verbose mode to assist users as they run through the workflow. | Same as the minimal, but in addition to being available for download, the analytical framework is also accessible as a free web-based interface with an easy-to-use GUI (*e.g.,* processed genotyping data are uploaded and results, including any QC, are returned). Software is actively maintained. Comprehensive documentation, tutorials, and examples are available to guide users through each step, and an intuitive workflow is included to ensure users can quickly begin using the framework. French and Portuguese instructions are available. | |
| Structure | The framework has a transparent and easy-to-follow structure. | Same as the minimal version, but with a modular structure allowing individual execution of each component of the overall analysis pipeline. It includes a developer manual to facilitate extensions and modification adjustments. | |
| Components | An algorithm to characterize (provide a categorical classification or probability estimate of) recrudescence or new infection status in recurrent samples. | Same as minimal, with the addition of inference of the study-level corrected efficacy rate. Additional optimal components also include summary outputs, visualization tools for the data and outputs, and testing datasets used for tutorials and validation. | In the minimal case, individual recrudescence or new infection calls can be analyzed in a subsequent, separate workflow to generate corrected efficacy estimates (for example, Kaplan-Meier survival analysis). Integrated analysis of efficacy may provide more accessible, reproducible, and less biased estimates. |

*(Continued)*

| Characteristic | Minimal Requirements | Optimal Requirements | Comments |
|---|---|---|---|
| Input type | Processed allele call data generated by a typical genotyping assay used for TES (*e.g.*, length-polymorphic loci, microsatellites, or amplicon sequencing results) for participants with recurrent parasitemia. Missing data at any level (*e.g.*, for a given locus within a sample, or for an entire sample) are accepted. | The framework can accommodate several genomic data types. The framework can also incorporate additional information that can be used to model uncertainty around allele calls, for example likelihood of false positive alleles from the allele-calling pipeline, parasite density or per-locus probability that a clone evades detection, complexity of infection, or data from intermediate time points. Additional non-genetic information that can be used to call recurrence, *e.g.*, day of failure/ time-to-event analysis, and *a priori* information on likely failure rates based on previous studies. Outcome data for all participants, not just those with recurrent parasitemia, to allow inference of the study-level corrected efficacy rate. | |
| Input format | Input data follows a format and structure clearly defined in the framework guidelines/ tutorials. | Input data are in a universally recognized format for processed genotyping data that are made publicly available for external validation of results. | |
| Primary Output | Characterisation (yes/no classification or probability estimate) of recurrent parasitemia as recrudescence or new infection. Clear information is provided about what the different outputs are and where they are stored. If genetic data are uninformative (*e.g.*, no genetic diversity/ all data are missing), the output is impartial (*e.g.*, the software refrains from classification/ returns the prior probability). | Same as minimal, with the addition of inference of the study-level efficacy/failure rate estimate, with measures of uncertainty for the estimated corrected efficacy at different timepoints that account for censoring. | If recurrences are characterized using probability estimates, the probability estimates may be used to propagate classification uncertainty into efficacy estimation |
| Secondary Output | In addition to individual classifications, the tool can also output additional summaries of the data; for example: - Any observed matches (alleles considered the same) between paired samples for each locus - Estimated or observed COI for each sample - Allele composition and diversity for all loci | Same as minimal, plus: - Probability of recrudescence for each pair - Estimated parameters (*e.g.*, sensitivity of allele detection) - Distributions of genetic distances between samples | Assessment of genetic distances between pairs of samples aids in evaluating the strength of evidence. The distribution between pairs where samples are taken from different individuals can be used to construct a hypothesis test under a not-recrudescence null [cite Plucinski & Barratt 2021]. Secondary outputs may be of utility beyond direct interpretation of TES results. TES failure rate estimates for sensitivity analyses (different allele frequency input) and consistency analyses (when analysis is based on 1,2,3..n loci, where n is the number of loci available. Failure rate estimates should ideally stabilize after several loci are analyzed, proving that estimates are not critically dependent on the number of loci. See Figure 2 in [8]. |

*(Continued)*

**Table 5.** (Continued)

| Characteristic | Minimal Requirements | Optimal Requirements | Comments |
|---|---|---|---|
| Scalability | The framework should handle various sample sizes, from small datasets to large-scale studies, and datasets of varied complexity, from low-complexity infections using data from few loci to high complexity infections with data from a large number of loci, with benchmarks for processing times. | Same as minimal, with effective parallelization (if needed) for very large datasets. | |
| Hardware requirements | The framework should be optimized for compatibility with standard computing devices commonly available to most users (*i.e.,* not require specialized hardware). | Same as minimal, with option of remotely hosted version on a virtual machine that can be freely accessed and used with only a web browser | |
| Reproducibility | External experts should be able to reproduce the same classification results using the genotyping data and data analysis scripts provided by the users | Same as minimal, and also including the ability of an external expert to reproduce the final efficacy estimates using the genotyping data, outcome data on all study participants, and analysis scripts provided by users. | Where appropriate, users should make all genotyping data and analysis scripts publicly available to allow external validation of results. For methods using stochastic algorithms, such as Markov Chain Monte Carlo, developers should specify a workflow users can follow to ensure they use an appropriate number of iterations per-chain and number of chains for reliable inference. Moreover, the workflow should record any seed(s) used to generate random numbers. |

**Table 6. Characteristics of the statistical algorithm for the TES data analysis framework.**

| Characteristic | Minimal Requirements | Optimal Requirements | Comments |
|---|---|---|---|
| Sensitivity and specificity | Account for following sources of error:<br>- Genotyping errors, including nondetection of minority alleles (false negatives), miscalling of alleles, and artifactual alleles (false positives)<br>- Chance matching of alleles in unrelated infections | Same as minimal | |
| Accuracy | Characterisation of recurrent parasitemia should be unbiased (*i.e.,* not be biased towards either recrudescence or new infection) by factors that can be accounted for statistically, *e.g.,* genetic diversity, polyclonal infections, undetected alleles, and other genotyping errors. The method allows for missing data, *i.e.,* failed genotyping at one or more loci in a given sample. | The tool also includes the ability to estimate corrected efficacy rates based on data from all individuals and an underlying relatedness structure in the local parasite population with incorporated survival analysis to account for right censoring. | |
| Consistency | Characterisation of recurrent parasitemia, either categorically or with a probability, as recrudescence or non-recrudescence. Should become more accurate and provide a stable output as the number of input loci increases. | Minimal plus the algorithm performs internal cross-validation to confirm that the results are robust to sub-sampling of loci. | Internal validity requires that as data from more loci are included, confidence in the classification increases. |

## Discussion

Accurate monitoring of antimalarial efficacy from TESs depends on accurate genotype correction stemming high quality molecular analysis and appropriate data analysis. This is particularly true in areas with high malaria transmission, where the likelihood of new infections after therapy is the highest. Importantly, increasing antimalarial resistance is being observed across multiple high-transmission areas of Africa [2,23–25]. Without methodologic standardization

across different settings and studies, it is unclear how to best interpret TES results to inform clinical care and practice. Ensuring methods meet the criteria outlined in the TPPs described here will help improve accuracy and consistency of genotype-corrected trial estimates while also providing appropriate uncertainty around these estimates, necessary to interpret and compare results over time and place. Further, it is essential that laboratory assays and the data analysis methods used to interpret results from these assays are compatible with each other. This includes model assumptions (*e.g.*, genotyping errors modeled by analysis methods should be appropriate for the type of data generated) and more practical considerations (*e.g.*, data output on allele calls can be input directly into analysis software, ideally facilitated by common data standards) [26]. Any laboratory and data analysis methods used should be systematically validated and its quality publicly documented so that clinical trial results, which have major policy implications, can be trusted. With appropriate design, these two components can provide a seamless evaluation workflow for those implementing TESs to monitor antimalarials- from genotyping methods to data analysis, and finally to drug efficacy estimates - making it straightforward to obtain and report transparent, reproducible, and accurate findings.

The last formal update to genotype correction guidance was published in 2008, with an informal update in 2021 [12,13]. The more recent update aimed to balance the state of science, favoring deep sequencing, and the feasibility of operationalization in malaria endemic settings. Since then, the expanded reach of deep sequencing in malaria endemic settings has greatly improved the context of what is feasible [27]. At the same time, recent TESs in high-transmission settings in sub-Saharan Africa have urgently highlighted the weaknesses of the older methods, yielding potentially inaccurate results that have complicated interpretation of the TES, delaying action and potentially contributing to the continued unabated spread of resistant parasites [28–33].

Key requirements for laboratory methods include a thorough validation process assessing parameters such as sensitivity in detection of minority alleles from several genetically diverse loci. Failing to detect alleles can result in imperfect detection of recrudescent parasites and underestimation of failure; too few loci or too low diversity in loci can result in successfully treated participants appearing to have recrudescence and overestimation of failure. Either of these issues can have a substantial effect on the final results, and both are potentially exacerbated in areas with high transmission where COI is often high and recurrent parasitemia is common. In areas of very low transmission, a lack of population genetic diversity can also present difficulties for genotyping classification. No laboratory method is perfect, and data analysis methods should account for these limitations and sources of error to the extent possible. However, major advances in sequencing technology and *Plasmodium*-specific targeted sequencing assays have dramatically improved the accuracy and decreased the cost of sequencing-based genotyping in the 27 years since genotype correction was originally proposed [4,34–39]. Importantly, high throughput sequencing is now available in national public health labs of over 80% of countries in sub-Saharan Africa, several of which are performing *Plasmodium* amplicon sequencing at scale [27]. While it is possible that older methods based on length polymorphisms may meet the criteria listed in the TPPs, it is likely that sequence-based methods will provide more accurate, reproducible, and cost-effective results, particularly if a large number of loci are required and/or sequencing of markers of drug resistance is also desired [10,39–41]. Moreover, the interpretation of length polymorphic markers analyzed by capillary sequencing is time consuming, difficult, and highly subjective, especially in non-experienced laboratories, limiting the reproducibility of those assays [10]. Sequence-based methods can produce data for a large number of loci simultaneously, making it easier to obtain results in a timely manner [39,42,43]. Leveraging existing sequencing platforms may also support overall usage and help build local expertise. Whatever methods are used, it will be important for labs performing assays to have qualified staff and reliable access to affordable reagents, external quality assurance, support for quality control including standardized controls, and appropriate training resources.

Transforming data from laboratory assays into trusted results requires that estimates produced by data analysis account for sources of genotyping error and uncertainty around classification estimates. Traditionally used data analysis methods based on match-counting do neither, are not easily transferable to different genotyping methods, and can provide

inconsistent, biased results [20,44]. Important considerations for analysis methods, as outlined in the TPPs described here, are that they account for statistical factors such as the expected number of alleles matching by chance in paired samples by taking into account population allele frequencies and COI. Technical factors, such as genotyping errors, also need to be accounted for. If done properly, results should be consistent regardless of the genotyping method used, though more accurate, high-resolution genotyping data should produce more precise estimates. To date, four statistical methods have been proposed, three of which have accompanying software packages [14,15,20,21]. Briefly, three of these methods are statistical inference approaches where likelihoods are derived from functions of the frequencies of observed alleles, and one method is a non-parametric, statistical threshold-based approach where the empiric null distribution of genetic distances is derived from pairs of unrelated Day 0 samples. These software packages may be considered for implementation after appropriate, ideally standardized, benchmarking is performed. Analogous to ongoing efforts for other data analysis tools, benchmarking will enable their full appraisal relative to the data analytic TPP and set a baseline for future method development [26]. Accessible implementation of the data analysis method will be important to ensure transparent, reproducible results. For example, fast, easy-to-use genotype correction software available locally and/or via a cloud-based platform would lower the barrier to high quality data analysis and encourage uptake. Even better would be extending this to a modular, integrated workflow for allele calling, genotype correction, study-level inference, and related analyses like associations between treatment failure and molecular markers of antimalarial resistance.

Validation of existing and newly developed methods to generate laboratory data and analyze these data for genotype correction should be evaluated in the context of the TPP framework described, with compliance assessed and documented. For example, sensitivity of the laboratory method for detecting minority alleles can be assessed via standardized controls containing mixtures of *P. falciparum* clones at known proportions; these would ideally be made available by a central facility and used in all laboratories genotyping samples from TESs. Likewise, sensitivity, specificity, and consistency of data analysis methods would benefit from *in silico* experiments using standardized datasets. Methods that do not meet minimum criteria should be avoided, and if used (*e.g.*, due to insurmountable logistical constraints), investigators should interpret final efficacy measures in light of the potential biases introduced.

Estimates of allele frequencies for loci used in genotype correction are important to ensure the accurate analysis of results. These estimates are ideally calculated using all (or, for large studies, a random sample of) Day 0 timepoints, from individuals with and without recurrent parasitemia, if resources allow. Further, *in silico* simulation experiments or bootstrapping of previous datasets can guide determination of how many total samples would need to be genotyped to provide enough precision for allele frequencies that allow for accurate estimation of recrudescence versus new infection status. Other options include incorporating data on allele frequencies from external sources, such as from recent or concurrent molecular studies from nearby sites.

Although we focus on TESs, these guidelines are meant to apply to any study or surveillance activity where antimalarial therapeutic efficacy is a primary outcome and where participants are subject to continuing risk of new infection during the period of follow up. Whereas many TESs are specifically designed to be incorporated as part of routine surveillance, clinical trials of new and candidate drugs typically have larger sample sizes, better resources, and require a higher standard of evidence. As such, clinical trials should, where possible, strive to meet the optimal criteria for all genotyping and statistical analysis components. Similarly, this manuscript focuses specifically on efficacy trials for participants with *P. falciparum*, as this is the species with the most documented resistance and is responsible for the majority of global burden. Evaluating antimalarial efficacy against other species is additionally complicated by differences in parasite lifecycle. For example, recurrence in patients with *P. vivax* can be due to either recrudescence, new infection, or relapse. The TPPs outlined here are a good starting point for development of *P. vivax*-specific guidance, but additional considerations are needed to design and validate genotyping and statistical approaches to molecular correction for *P. vivax* trials.

As outlined above, some laboratory and data analysis methods have already been designed to account for factors such as genetic diversity, polyclonal infections, undetected alleles, and other genotyping errors. However, there are additional factors

that may affect the accuracy of genotype correction, including parasite population structure, latent liver stages at enrollment, completely undetected parasites (*e.g.*, due to organ tissue sequestration of parasites), and persistent gametocytes, which are not easily distinguished from asexual parasites by DNA genotyping [4,13]. Some of these factors may distort genotype correction at the per-participant level. Fortunately, per-participant precision, while important for detecting associations with treatment failure (such as parasite mutations or plasma drug levels) is not the primary goal of a TES. At the study level, these issues can be mitigated by randomization in multi-arm studies. Single-arm studies are more vulnerable to classification bias, especially if relying on unvalidated genotype correction methods. However, this bias may be addressed using sensitivity analyses or models that go beyond the standard capabilities of current genotype correction software. Some of these advanced capabilities could be integrated into future data analysis tools, likely requiring a revision of the current TPP.

Exclusive use of fragment-length polymorphic loci and simple heuristic-based data analysis methods are no longer supported by the current state of science. Where possible, newer techniques that have been benchmarked and appraised as a core part of TESs should become the standard. It may be of interest to perform traditional genotyping methods in parallel for some studies, to facilitate evaluation of temporal trends including prior studies and provide direct comparison of results from different methods. Evidence that newer methods provide different and (based on rigorous validation) more accurate results should further motivate switching to modern laboratory and data analysis tools and to using their results for the primary analysis of trials and policy decision making.

At this point, it is not clear if the scientific community is well-advised to coalesce around a single set of loci, laboratory assay, and data analytic method that will replace the length-polymorphic markers and match counting from the 2008 formal guidance and 2021 informal update. Fortunately, it is likely that several genotyping and data analysis method combinations will be able to meet the criteria outlined here in the near future, and thus that multiple independent approaches will yield high-quality and similar results. Importantly, one of the key concepts emphasized in this TPP is consistent reporting of uncertainty. If uncertainty is properly measured and reported, results using different laboratory and data analytic methods will more likely be comparable. Consistency in results is critical and will require validation, *e.g.*, via *in silico* simulations, standardized reference samples, and parallel application in real TESs. Moreover, as methods become more sensitive and discriminatory, results should be more robust and stable across different methods. Adherence to the best practices for new genotype correction approaches over the coming years will help build the evidence base around the comparability and robustness of different approaches, setting the stage for the next phase of antimalarial efficacy monitoring.

## Supporting information

**S1 Table. Issues negatively affecting accuracy of TES genotyping classification using currently recommended methods.** All issues listed here have a greater effect on efficacy outcomes as the proportion of study participants with recurrent infection increases (i.e., when transmission and/or recurrence rates are high).
(DOCX)

**S2 Table. Guidelines for number and diversity of loci required for accurate genotype correction.** Minimal heterozygosity* per locus needed to achieve the classification performance of 95% specificity and 95% sensitivity determined by simulations and classification using a statistical method that accounts for complexity of infection (COI), allele frequency, and imperfect detection of minority alleles.<insert ref> For simplicity, the current simulation assumes both samples have the same COI. NA are present when a heterozygosity of 0.95 in all loci is not high enough to achieve 95% sensitivity and specificity. *Heterozygosity is a measure of allelic diversity at the locus. It is defined as the probability that two alleles taken at random from the local population are different, *i.e.*, higher heterozygosity indicates higher diversity.
(DOCX)

**S1 Code. The R code for the simulations used to generate S2 Table can be found here.**
(R)

## Acknowledgments

We thank Sam Jones for his insightful comments on this manuscript.

## Author contributions

**Conceptualization:** Mateusz M. Plucinski, Amy Wesolowski, Daniel Neafsey, Christian Nsanzabana, Bryan Greenhouse.

**Formal analysis:** Inna Gerlovina.

**Funding acquisition:** Bryan Greenhouse.

**Writing – original draft:** Mateusz M. Plucinski, Amy Wesolowski, Inna Gerlovina, Aimee R. Taylor, Jessica Briggs, Andrés Aranda-Díaz, Monica Golumbeanu, Marko Bajic, Christian Nsanzabana, Bryan Greenhouse.

**Writing – review & editing:** Mateusz M. Plucinski, Amy Wesolowski, Inna Gerlovina, Aimee R. Taylor, Jessica Briggs, Andrés Aranda-Díaz, Monica Golumbeanu, Jeffrey A. Bailey, Joel L. N. Barratt, Caroline Buckee, Awa B. Deme, Ingrid Felger, Anita Ghansah, Ian Hastings, Johanna Helena Kattenberg, Alfredo Mayor, Didier Menard, Leah F. Moriarty, Daniel Neafsey, Lucy Okell, Isabella Oyier, Jaishree Raman, Philip J. Rosenthal, Anna Rosanas-Urgell, Robert Verity, Sarah K. Volkman, Christian Nsanzabana, Bryan Greenhouse.

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
