## [Decision Letter · Decision Letter 0]

25 Mar 2026

PGPH-D-25-03787

Target product profiles of laboratory and data analytical frameworks for genotyping to monitor antimalarial efficacy

Dear Dr. Greenhouse,

Thank you for submitting your manuscript to PLOS Global Public Health. After careful consideration, we feel that it has merit but does not fully meet PLOS Global Public Health’s publication criteria as it currently stands. Therefore, we invite you to submit a revised version of the manuscript that addresses the points raised during the review process.

We look forward to receiving your revised manuscript.

Kind regards,

Xin Hui Chan

Academic Editor

Journal Requirements:

1. Your submission has a current Financial Disclosure statement. However, there is no funding information on the submission form indicates that you received funding. Please indicate by return email the full and correct funding information for your study and confirm the order in which funding contributions should appear. Please be sure to indicate whether the funders played any role in the study design, data collection and analysis, decision to publish, or preparation of the manuscript.

Reviewers' comments:

Reviewer's Responses to Questions

**Comments to the Author**

1. Does this manuscript meet PLOS Global Public Health’s publication criteria? Is the manuscript technically sound, and do the data support the conclusions? The manuscript must describe methodologically and ethically rigorous research with conclusions that are appropriately drawn based on the data presented.

Reviewer #1: Yes

Reviewer #2: Yes

2. Has the statistical analysis been performed appropriately and rigorously?

Reviewer #1: I don't know

Reviewer #2: Yes

3. Have the authors made all data underlying the findings in their manuscript fully available (please refer to the Data Availability Statement at the start of the manuscript PDF file)?

Reviewer #1: No

Reviewer #2: Yes

4. Is the manuscript presented in an intelligible fashion and written in standard English?

Reviewer #1: Yes

Reviewer #2: Yes

5. Review Comments to the Author

Reviewer #1: The manuscript focuses on guidelines for genotyping correction for therapeutic efficacy studies (TES). Genotyping for TES has been challenging due to the lack of accurate methods. Genotyping in TES is focused on the question of whether infections are new versus recrudescent, and therefore whether drug treatment has been effective. In recent years methods for genotyping have evolved and become more sensitive and accurate due to availability of polymorphic marker panels and next generation sequencing methdologies. In addition analytical methods have been developed that provide information on relatedness, which is critical to differentiating between new and recrudescent infections.

The manuscript describes the results of an expert review of methodologies and provides guidelines for applying these genotyping strategies to TES in the form of TPPs, Target Product Profiles. The evaluation included simulation of desired levels of diversity and the nos of markers required for different infection complexities, which are complicating aspects of malaria which can vary greatly at different transmission intensities. Both desired and essential characteristics are described.

This is a timely and important publication to ensure best practice/high quality research is used for decision making. Generally the manuscript is well-written and makes a strong rationale for the TPPs for TES genotyping tools that are agnostic to the numerous genotyping panels, developed for Plasmodium. The simulation is appropriate and the TPPs are extensive and cover a large range of required characteristics. They were developed/refined by a large list of experts that this reviewer confirms are indeed global experts in malaria epidemiology and genotyping, some of whom have developed their own panels specifically for the use case described. Others are bioinformaticians/modellers familiar with the complexities and challenges of Plasmodium molecular data. While the study focuses on genotyping correction for TES, it makes an important note that these guidelines may be just as important for malaria studies and surveillance activities, including research. These guidelines could also be implemented outside of clinical trials and implemented in all research contexts.

Specific comments

1) The paper provides goals for genotyping tools to sensitively and specifically classify infections , but does not show examples of how the data may be used and is therefore not accessible to a broad audience. A figure/schematic would be useful to illustrate how the TPPs will work in practice.

2) The technical details in the main table will only be of use to those using/developing genotyping approaches – can a lay version of this table be produced to make the information more accessible to a broad audience including control programs – this will help to illuminate some of the key issues.

3) The authors make recommendation to use mixtures of P. falciparum clones at known proportions to identified sensitivity of assays but also validation of analysis methods – but do not give specific details on this. Suggest to include an example protocol/workflow.

4) No mention of Plasmodium vivax and the issues particular to clinical trials e.g. relapse vs recrudescence. Very challenging. Other species also not mentioned. The paper would benefit from at least a mention of the species relevant to their recommendations.

5) Line 342, mentions “weakness of older methods” – but authors do not describe nor discuss what these are.

6) Advice provided that sequencing, i.e., amplicon-based, would be more accurate and cost effective compared to older methods and capillary designed experiments etc however no specific information was provided on this and evaluations of NGS vs capillary electrophoresis.

7) Line 384, “To date, four statistical methods have been proposed, three of which have accompanying software packages.(14,15,20,21)”. . This paragraph assumeds knowledge of these methods and packages so it would be useful to include named and brief details of the packages.

8) Limited details of the simulation methods or underlying data provided

Reviewer #2: Manuscript:

Target product profiles of laboratory and data analytical frameworks for genotyping to monitor antimalarial efficacy

This manuscript addresses an important topic related to improving the accuracy and reliability of genotyping approaches used in Therapeutic Efficacy Studies (TESs) for antimalarial drugs, specifically for Plasmodium falciparum. Strengthening the framework used to distinguish recrudescence from reinfection is critical for accurate treatment efficacy and policy decisions. The manuscript provides guidelines for laboratory and analytical methods aimed at standardising genotyping practices.

It appears that these guidelines were generated from meetings across collaborators in US, who constitute most of the authors. While the work reflects considerable expertise in assay development and statistical analysis, it is surprising that representation from endemic regions is limited. Only a few contributors from Africa are included, and none from Asia, where monoclonal infections present additional challenges for TES. Broader input from endemic settings and independent research groups would strengthen the manuscript’s relevance.

Areas where the manuscript could be improved:

Geographic representation and epidemiological considerations: The manuscript briefly mentions other geographic and epidemiological challenges, but a more thorough discussion of the implications for monoclonal versus polyclonal infections in different regions, and situations of outbreaks or (re) introductions would strengthen the work.

Practical pipeline guidance: The manuscript would benefit from a more pipeline-oriented approach that guides users in practical decisions based on laboratory capacity (e.g., MinION vs. Illumina, capillary), regional transmission intensity, and other situational factors. Including flowcharts or decision tables would make the guidance more actionable and improve readability.

Tables: Given the number of tables, the manuscript can become difficult to navigate. Some tables are lengthy and include information that may be obvious to the target audience. For example, Table 1 could be condensed into a few sentences. Tables 3 and 5 contain information that is not critical and could be trimmed. Simplifying tables and combining them with workflow illustrations could improve usability and readability.

Economic considerations: Including example costs for personnel, equipment, and consumables across platforms in representative endemic countries would provide a practical context for implementation.

This work has strong potential to contribute to the field by guiding the development of robust, standardised TES genotyping pipelines. With additional attention to geographic representation, methodological limitations and cost, and practical implementation, the manuscript could provide a solid foundation for establishing community standards.

6. PLOS authors have the option to publish the peer review history of their article (what does this mean?). If published, this will include your full peer review and any attached files.

**Do you want your identity to be public for this peer review?** For information about this choice, including consent withdrawal, please see our Privacy Policy.

Reviewer #1: No

Reviewer #2: No

Figure Resubmissions:

---

## [Editor Report · Decision Letter 1]

4 May 2026

Target product profiles of laboratory and data analytical frameworks for genotyping to monitor antimalarial efficacy

PGPH-D-25-03787R1

Dear Dr. Greenhouse,

We are pleased to inform you that your manuscript 'Target product profiles of laboratory and data analytical frameworks for genotyping to monitor antimalarial efficacy' has been provisionally accepted for publication in PLOS Global Public Health.

Best regards,

Xin Hui Chan

Academic Editor